



# Evaluation of Obstacle Modelling Approaches for Resource Assessment and Small Wind Turbine Siting: Case Study in the Northern Netherlands

Caleb Phillips[1], Lindsay M. Sheridan[2], Patrick Conry[3], Dimitrios K. Fytanidis[4], Dmitry Duplyakin[1], Sagi Zisman[1], Nicolas Duboc[3], Matt Nelson[3], Rao Kotamarthi[4], Rod Linn[3], Marc Broersma[5], Timo Spijkerboer[5], Heidi Tinnesand[1]

[1]National Renewable Energy Laboratory, Golden, CO, USA
[2]Pacific Northwest National Laboratory, Richland, WA, USA
[3]Los Alamos National Laboratory, Los Alamos, NM, USA
[4]Argonne National Laboratory, Argonne, IL, USA
[5]EAZ Wind, Rijswijk, Netherlands

*Correspondence to*: Caleb Phillips (caleb.phillips@nrel.gov)

**Abstract.** Growth in adoption of distributed wind turbines for energy generation is significantly impacted by challenges associated with siting and accurate estimation of the wind resource. Small turbines, at hub heights of 40m or less, are greatly impacted by terrestrial obstacles such as built structures and vegetation that can cause complex wake effects. While some progress in high-fidelity complex fluid dynamics (CFD) models has increased the potential accuracy for modelling the impacts

5 of obstacles on turbulent wind flow, these models are too computationally expensive for practical siting and resource assessment applications. To understand the efficacy of available models *in situ*, this study evaluates classical and commonly used methods alongside new state-of-the-art lower-order models derived from CFD simulations and machine learning approaches. The evaluation is conducted using a subset of an extensive original dataset of measurements from more than 300 operational wind turbines in the northern Netherlands. We find that data driven methods (e.g., machine learning and statistical

10 modelling) are most effective at predicting production at real sites with average error in annual energy production of 2.5%. When sufficient data may not be available *de novo* to support these data-driven approaches, models derived from high fidelity simulations show promise and reliably outperform classical methods. On average these models have 6.3-11.5% error compared to 26% for classical methods and 27% baseline error for reanalysis data without obstacle correction. While more performant on average, these methods are also sensitive to the quality of obstacle descriptions and reanalysis inputs.

## 1 Introduction

15 Distributed wind (DW) energy constitutes a small but growing market with a total global installed capacity estimated at approximately 1.8 GW (Orrell et al., 2021). As compared to their utility-scale counterparts, small wind turbines provide significant opportunity to diversify energy production and address a market niche not otherwise served, since they are



particularly well-suited to industrial, agricultural, and campus-level installations. Despite their significant potential as a component of a larger distributed energy resource (DER) portfolio, adoption of small wind is practically hindered by a number of challenges including project costs, availability of incentives, and confidence in the underlying technologies. Confidence, in particular, is strongly impacted by the practical accuracy in the prediction of energy production at a site of development interest (Fields et al., 2016). Accurate energy production estimates can only be obtained by predicting the available wind resource. While large-scale wind plants can afford to perform detailed observation-driven wind resource and site assessments, small-scale wind projects typically rely on less-expensive use models and assessment tools to survey the energy production potential due to lower financial scope of anticipated projects. While the dependence on models is greater, the challenge to model the resource accurately is also significantly higher: small wind turbines are more greatly impacted by turbulence and wake effects from surrounding terrestrial obstacles such as buildings, trees, and vegetation (Drew et al., 2015).

The state of the art in assessment of obstacle influence on downwind wind fields for small-wind siting and resource assessment is characterized by the usage of tools that provide a mixture of analytical modelling and heuristic estimation (Poudel 2019). The effect of isolated obstacles on the flow field structure has been studied in the past both experimentally (Schofield 1990; Martinuzzi 1993; Snyder 1994; Hussein 1996) and numerically (among others Lakehal 1997; Krajnovic 1999; Yakhot 2006a and 2006b). Velocity data from such studies can be used to approximate the effect of buildings on wind velocity profiles in the wake of isolated obstacles. Analytical models from the literature (Robins 2020, Counihan 1974, Kothari 1979, and Peterka 1985) have also been developed for the case of isolated cubes with zero wind angle of attack (conditions where wind approached the building perpendicular to one of its sides). Additionally, Perera (1998) used data from wind tunnel experiments and developed a model suitable to predict the velocity deficit at the wake of an infinitely long obstacle when the winds are perpendicular to the length of the obastacle.

As a potential compromise between high-fidelity solutions that are too-computationally intense for non-expert use, such as Large-Eddy Simulations (LES; Castro 2017; Bieringer 2021), and fitted models derived from isolated experimental campaigns, lower-fidelity computational fluid dynamics (CFD) models have been applied to more complex obstacle geometries to solve flow patterns in urban areas (Tominaga and Stathopoulos 2013). Among these are Reynolds-averaged Navier-Stokes (RANS; Bruse 1998; Gowardhan 2011) modelling approaches, which while computationally faster, do not capture the physics at all scales relevant for urban flow and turbulence modelling. While there have been recent demonstrations (Bierenger 2021) of LES running ~100 times faster when using graphical processing unit (GPU) approaches than when using the traditional central processing unit (CPU), the expertise required to set-up such models and relatively long computational run times for even these lower-fidelity CFD models are not feasible for operational use (Hertwig 2018; Tominaga 2016).

In this study, we evaluate practical methods for modelling the impact of obstacles on siting and resource assessment, selecting those models from the literature that are sufficiently performant and usable for this application, specifically (1) the Perera and SHELTER (WaSP) models (Section 3.1) (2) a pair of new models developed using data from novel CFD simulations (Section 3.2) (2) a modified QUIC-URB model adapted from urban dispersion modelling (Section 3.3), and (4) custom fitted machine learning models using site-specific data (Section 3.4). To evaluate this array of models we leverage a comprehensive



dataset from the northern Netherlands combining meteorological tower measurements and turbine production data for more than 300 turbines over multiple years covering a large geographic area. To our knowledge this is the first study of its kind and provides an original benchmark for understanding the accuracy of practical resource assessment and siting methods in the

distributed wind context. Acknowledging that obstacle assessment is only one step in the process of siting, we attempt to understand the scale of errors associated with other components as well, including (a) baseline reanalysis estimates of the mesoscale wind resource (b) vertical and spatial interpolation necessary to map reanalysis data to a specific site and target turbine height (c) bias correction of reanalysis data using regional measurements and (d) the accuracy and completeness of obstacle descriptions.

60        This paper is organized as follows: in the next section (Data) we describe and characterize the measurement and model data used in this study. In Section 3 (Methods) we describe the models evaluated and the experimental design and metrics used for validation. Section 4 (Results) provides results from the validation exercise, and Section 5 (Conclusions) concludes with discussion of results, limitations inherent to this study, and areas of potential future work.

65                                       **Table 1.** Summary of data sources

| Data Source | Variables | Locations | Data Points | Temporal Resolution | Duration |
|---|---|---|---|---|---|
| EAZ Turbine Production | Power (kW) | 327 | 5,118,243 | Hour | 1 to 4 years (depending on turbine installation date) |
| EAZ Meteorological Tower | Wind speed (m/s), Wind direction @ 15m | 1 | 6,514 | Hour (down-sampled from 100ms) | 8 months |
| KNW-Atlas Reanalysis Dataset | Wind speed (m/s), Wind direction, Air density | 2.5km Spatial Resolution for Study Area | N/A | Hour | 20 years (1998-2019) |
| AHN3 Lidar Digital Surface Model (DSM) | AGL Terrain/Height and Elevation (m) | 0.5m Spatial Resolution for Validation Locations | N/A | N/A | 2014-2019 |
| 3DBuildings.com Vector Building Data | Polygonal Buildings and Height (m) | Approx. 53km x 49km area near the city of Groningen, Netherlands | 22,535 buildings | N/A | Downloaded between Dec 2020 and Jun 2021 |

## 2 Data

Table 1 provides a brief summary of the datasets used in this study, in the following subsections we will discuss each in detail.



## 2.1 Production Data from Turbines

To form a basis for evaluating model performance, we have made use of an extensive dataset of measurements from EAZ
Wind, a turbine installer and operator in the northern Netherlands. Figure 1 shows the locations of these turbines as well as an
example site used in this study. In addition to these turbines, we utilize data from one meteorological tower installed for
International Electrotechnical Commission (IEC) validation. For this tower, wind speed and direction are determined with a
calibrated, tower-mounted anemometer. For each turbine, power production data sampled hourly is used to approximate wind

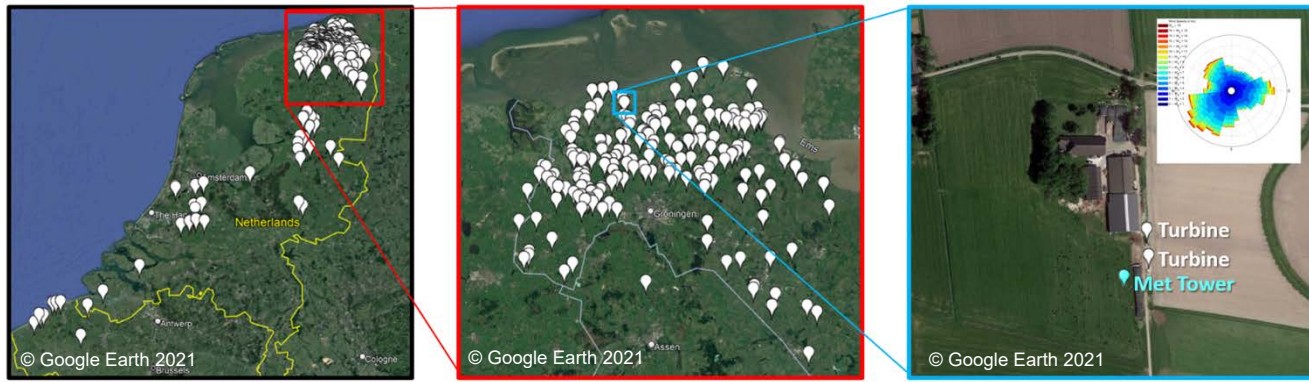

**Figure 1.** Production data were drawn from more than 300 turbines located in the northern Netherlands. Each turbine has a 15m hub
height and provides power production data hourly. The area of detail includes the majority of turbines which are the focus of our obstacle
modelling effort. An example site is given in the bottom right of the figure, showing the placement of two specific validation turbines
(numbered 107 and 108), the IEC meteorological tower and wind rose for this location. When the wind direction is north/northwest, these
turbines are in the wakes of the buildings and trees on the site while other wind directions are largely free of obstructions.

speed using a manufacturer provided power curve. All turbines used in this study are EAZ-Twelve with a standardized hub
height of 15 meters.

   When converting power to assumed wind speed, we are unable to differentiate approximate wind speed for those
times when the power production is zero, when the turbine may be curtailed by the grid or offline for maintenance. We also
cannot differentiate between higher wind speeds when the turbine is operating at rated power. For the remaining cases where
the power generation is between 0 and 11 kW, the power curve is invertible, allowing a simple computation of windspeed
through interpolation. Those turbines nearby the meteorological tower show good agreement between the anemometer
measured wind speeds and those inferred from power generation (RMSE: 0.98m/s, MAE: 0.63m/s, Mean Bias: -0.15m/s).
Some of the residual differences between the anemometer and the inferred speeds from power generation are likely due to
different wake impacts as the turbines are closer to and more in line with obstacles relative to the dominant wind flow directions
compared to the tower.



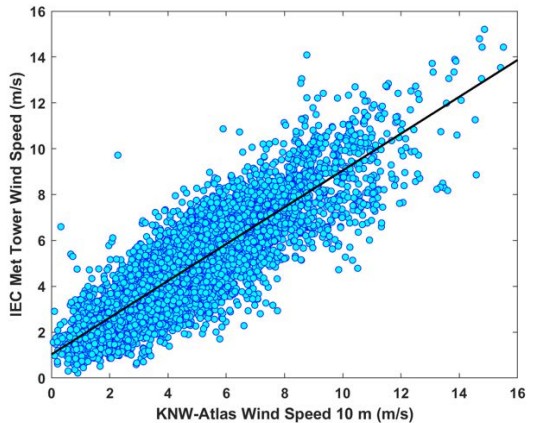 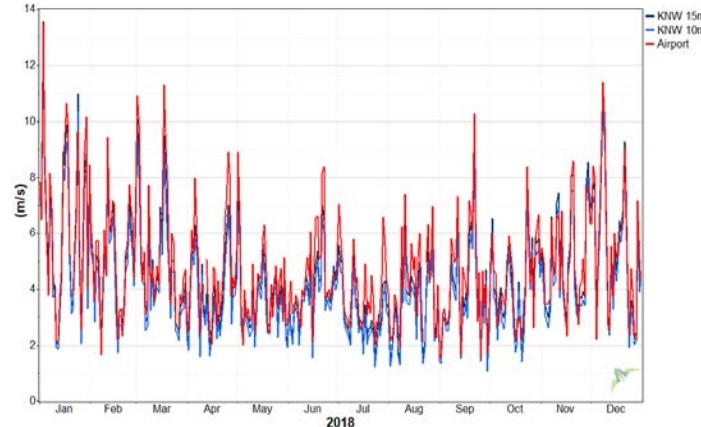

**Figure 2.** These two plots show the approximate accuracy of the KNW-Atlas reanalysis data in study region. The left figure compares reanalysis data at 10 m (no vertical interpolation) to those data collected from the IEC meteorological tower at 15m. A small negative bias is observed (-0.12 m/s) observed with an $R^2$ of 0.75. The right figure compares the KNW reanalysis data product to data from the Amsterdam Schipol airport in 2018 using the nearest KNW site (14 km distant).

## 2.2 Reanalysis Mesoscale Atmospheric Data

For inflow data to the models studied, we utilize mesoscale reanalysis data from the KNW-Atlas dataset whose underlying model is ERA5-Interim (Wijnant 2015). Associated validation studies found that KNW-Atlas overestimates the 10-m winds by 0.3-0.4 m/s using satellite-derived offshore winds for most of the North Sea and underestimates the 10-m winds by 0.1-0.3 m/s near the Dutch coastline (Stepek 2015). Observations from one *onshore* met tower were used to correct the entire product. Due to the limited prior assessment of onshore accuracy, we performed our own assessment. Figure 2 shows the results of this validation, demonstrating agreement with IEC met tower and nearby airport measurements, further confirming the limited prior validation results.

In order to map the KNW-Atlas data to the location of each turbine for the purpose of validation, we utilized best-practices for vertical and spatial interpolation techniques, previously establishedd for 40-m hub heights in the United States (Duplyakin 2020). Specifically, we compared the inverse distance weighting (with 16 interpolation points) strategy for spatial interpolation and the linear interpolation strategy for vertical interpolation. According to our recent work, this combination produced wind estimates with the lowest validation errors, characterized by the Mean Absolute Error estimates obtained for 63 validation sites in the U.S. (154 site-height combinations) investigated in that study. Despite their relative efficacy in the U.S., we found that a simpler method utilizing nearest-neighbor spatial interpolation and a log law spatial interpolation was slightly better performing in the area of study. Hence, that approach is used here.

We also evaluated two techniques to perform additional region-level bias correction (BC) of the reanalysis data. BC Technique 1 (BC1; Turbine Data) utilizes wind speed data from each turbine, converted from power as discussed above to fit a multiple linear regression utilizing hour of the day, month of the year, wind direction, and reanalysis wind speed as predictor variables. To prevent under fitting due to the dearth of higher wind speeds in this dataset, we impute a constant 13 m/s when



the turbine is operating at rated power. BC Technique 2 (BC2; Met Data) utilizes data from the meteorological tower. While this data is more limited in spatial scope, it is higher resolution, providing greater accuracy. The addition of bias correction resulted in a reanalysis data product closer to observations, however, as both methods necessarily include some effects from obstacles, there may be double counting of obstacle impacts when combined with a separate obstacle model.

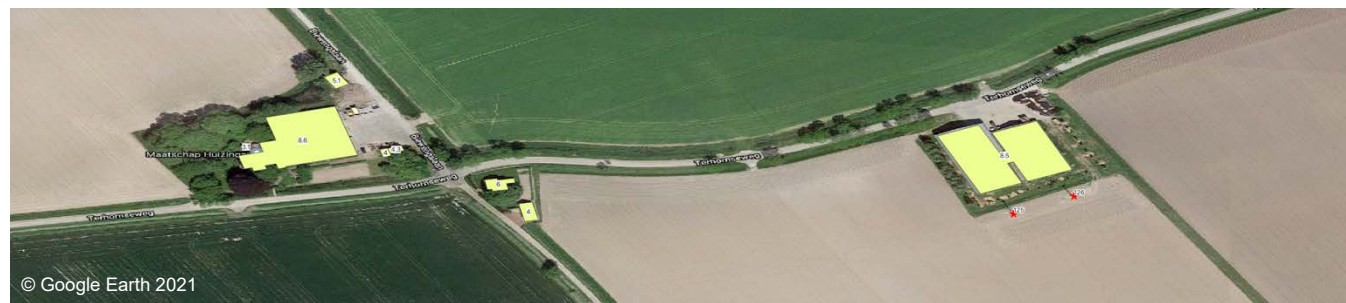


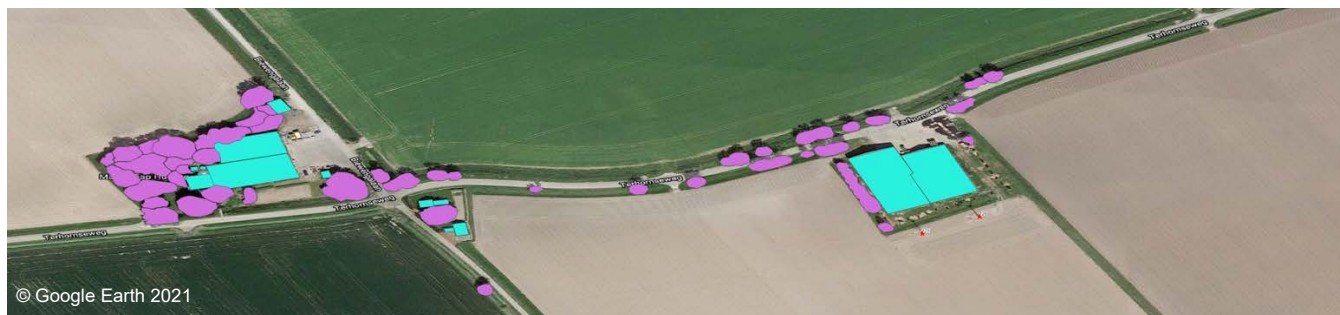

**Figure 3.** (a; Top) Obstacle annotations (yellow) for a site with two turbines using data commercially available from 3DBuildings.com (b; Bottom) Full annotations using semi-automated annotation process including both buildings and vegetation, enriched with lidar data. Vegetation is shown in violet and buildings in cyan.

**2.3 Obstacle Data**

To support obstacle assessment we developed a standardized method for mapping of obstacles—*buildings* or *vegetation*— around the studied turbine sites. At the end of this process, each of these records becomes an entry in a geospatial data frame, in which height estimates accompany the two dimensional polygons that represent obstacle shapes. This data frame can be "sliced" for analysis and visualization of buildings only, vegetation only, or the combination of the two obstacle types, as

illustrated later in this section.

We begin with information from 3DBuildings.com (2021) which provides a high-resolution building dataset at the inexpensive average rate of approximately $0.06 USD/km$^2$ and/or $0.07 USD/building for the selected area in northern Netherlands. Similar data can be obtained for other areas in the world, however the rates are likely to be different as they are directly tied to the level of urbanization in each area of interest. Figure 3(a) provides an example of the annotations available

commercially, which correctly identifies building obstacles.



In cases where the commercially available data may be incomplete, and to account for vegetation, which is not included in the commercially available data, we have developed a semi-automated process using publicly available lidar data from the AHN3 Lidar Digital Surface Model (DSM). To simplify annotation, we extract the DSM layer for the 1 km x 1 km square encompassing a given turbine. The DBSCAN clustering technique is used with a convex hull mapping algorithm to

determine preliminary obstacle polygons (Schubert 2017). We then proceed to manually correct the obstacles and use a visible satellite layer in the QGIS software to modify the polygons around buildings and trees separately. Once the polygons are extracted, we compute the heights of the obstacles by masking the underlying DSM data with the annotated obstacles and calculate the mean height for each. The resulting polygonal data are stored as a portable GeoJSON file for each turbine. Across the full set of validation sites, filtering out null height obstacles, and considering obstacles with heights greater than 1 meter,

257 buildings and 1353 trees/vegetation polygons were located. Building heights range from 1 to 12 meters while vegetation heights range from 1 to 15 meters.

## 3 Methods

### 3.1 Classical Models

The Perera model was developed in 1981 using wind tunnel measurements (Perera 1981). It provides a closed form equation for the velocity deficit behind a thin (in streamwise direction), infinite-length obstacle of arbitrary porosity, similar to a fence or hedgerow.  This model was designed for a scenario where the wind was perpendicular to the length of the obstacle. Despite its apparent limited applicability to other situations, it has found broad applications in commercial tools and remains well known in the small wind community (Poudel 2019). Besides the classic Perera model, various extensions exist including the

SHELTER model proposed as part of the WaSP toolkit (Alstrop 1999), which allows for limiting the obstacle length to better model buildings and other finite-length shapes. We have implemented these models following the descriptions in the literature and will refer to the finite-length version of Perera as Perera+.

One significant limitation of both models in their usage for wind turbine siting is that they do not provide reasonable values for obstacles nearer than 5*h (or 7*h per the original Perera paper), where h is the obstacle height. Figure 4 shows this

visually where the velocity deficit factor for different obstacle heights and measurement distances is given. The triangular region at left is the area for which Perera should conservatively abstain from making an estimate as this represents locations closer than 5*h. Since these nearby locations may incur the largest impact in terms of wake and turbulence on neighbouring turbines, there is significant concern that for DW siting and resource assessment applications the Perera method may underestimate obstacle impacts. Nevertheless, due to their popularity, we include these models as a baseline upon which to

compare subsequent models.

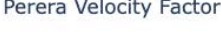

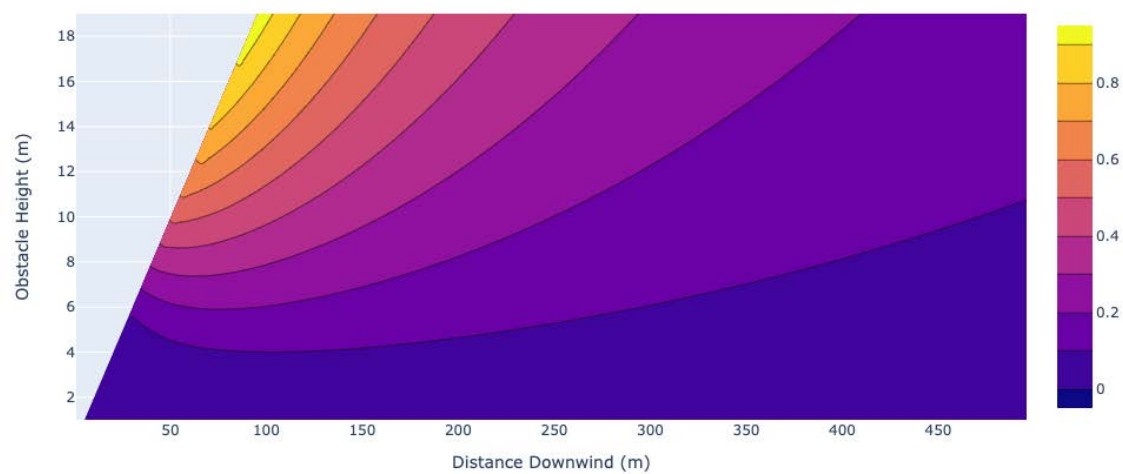

**Figure 4.** Velocity deficit factor prescribed by the Perera model for different combinations of obstacle heights and distances downwind. The missing data in the triangular section at left are those points within 5h where the model cannot make an accurate estimate.

### 3.2 ANL and ANL/ML Models

We developed several low-order models (LOM) that are designed to update the Perera model. An extensive dataset from RANS simulations for the prediction of the flow structure at the wake of buildings (Fytanidis et al. 2021a) were used to train the physics-informed/analytical data-driven model and the solely machine learning based model, referred herein as ANL and ANL-ML respectively. Specifically, the high-order spectral-element based solver NEK5000 (Fischer et al. 2008) was used to solve three-dimensional turbulent flow equations assuming incompressible flow using realistic boundary conditions to mimic

characteristics of real turbulent atmospheric boundary layer flows. The k-τ turbulence closure (Speziale et al. 1992) in combination with the Boussinesque approximation were used for the estimation of Reynolds stresses. The accuracy of the numerical results was evaluated against wind tunnel observations (Fytanidis et al. 2021a, 2021b, using data from Snyder 1995).

The validated NEK5000 computational results were used as training dataset for the evaluation of parameters in a physics-informed data-driven model (Fytanidis 2021b) that takes into consideration different angles of attack and the effect of

different building aspect ratios for the prediction of wake characteristics. Specifically, the dimensions of the enclosing cuboid (Figure 5a) aligned with the direction of the incoming velocity were used as input parameters for the training of the coefficient for the physics informed ANL model. The developed model is a new, generalized version able to predict the wake characteristics under various angles of attack and building aspect ratios. Additionally, a correction factor for the prediction of the acceleration due to the formation of horseshoe vortex around the building was developed (Fytanidis et al. 2021b). The low-

order model parameters for various angles of attack and building aspect ratios were estimated using the surrogate model



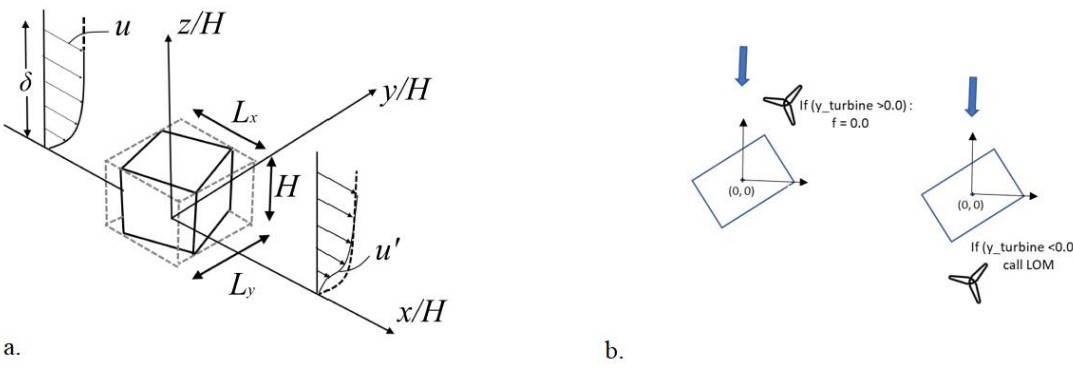

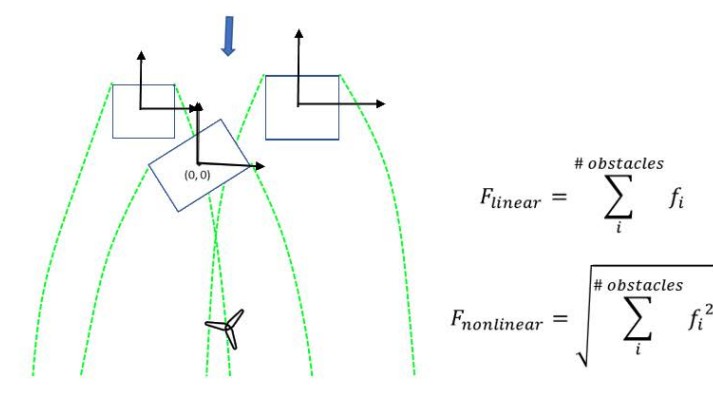

**Figure 5.** Schematic describing ANL and ANL/ML models.

technique combined with the machine learning based algorithms included in the open-source package Tensorflow (Abadi 2015). The applied Neural Network architecture consists of 4 branches for the main wake model component of the LOM and 3 branches for the horseshoe correction of the LOM.

The developed model uses a local coordinate system for each obstacle and evaluates the location of the turbine with

respect to the centre of the enclosing cuboid (Figure 5b). When the turbine is located in the wake of obstacle, the low-order model evaluates the velocity deficit $f_i$. If the turbine is located upwind of the building the velocity deficit equals zero ($f_i$=0). Finally, the solution for each obstacle were superimposed (Figure 5c) using a linear and a non-linear superposition technique (see more details in Lissaman 1979; Katic 1986; and Vogel 2020).



### 3.3 LANL/QUIC Model

Models developed for decision support related to transport and dispersion of pollution, chemical-biological-radiological agents, and smoke in population centres cannot be served by CFD approaches due to their computational requirements. Thus, fast-response models have existed for three decades (Rockle 1990) to serve these specialized applications. These tools fill an important gap in dispersion modelling between extremely rapid but overly simplified flat-earth analytical models typically employed by the emergency response community and the high-fidelity but computationally expensive CFD codes. One of

these tools is the Quick Urban & Industrial Complex (QUIC) dispersion modelling system, which was designed to compute wind fields in dense built-up urban areas using the diagnostic wind solver QUIC-URB (Brown et al. 2013). For dispersion modelling, these wind fields are then used by the Lagrangian random-walk model QUIC-PLUME to predict transport and dispersion of gas and particle releases in urban areas. Similarly, QUIC-URB or like-minded models (Kaplan and Dinar 1996; Wang et al. 2005) can provide wind fields for DW applications where small-scale wind energy developers do not have the

time, resources, or expertise to employ higher-fidelity CFD modelling tools. QUIC can run on a laptop with one simulation requiring only seconds to minutes and has been demonstrated to predict wind fields (Neophytou et al. 2011) and plume dispersion footprints (Hertwig et al. 2018) that were comparable to CFD modelling results for practical applications, especially considering the much shorter run times.

    The QUIC system's empirical diagnostic wind solver, QUIC-URB, is based on the concept developed by Röckle

(1990). The 3D mean wind field is initialized using one or more vertical profiles of wind speed and direction that can either be directly measured or an extrapolation from a single measurement point. The ambient or background velocities are determined in horizontal planes between all of the profile locations using a two-dimensional Barnes-mapping interpolation scheme (Pardyjak et al., 2004). QUIC-URB then uses empirical parametrizations to modify the initial wind field to account for building effects (Brown et al., 2013) and vegetative canopy drag (Nelson et al., 2009). The original building wake algorithm

as detailed in Röckle (1990) and Kaplan and Dinar (1996) divides the wake downwind of a building into two regions: (a) the recirculating cavity where the flow direction is reversed from the ambient flow and (b) the wake region that transitions the flow back to the undisturbed ambient conditions. The original wake region is defined by a quarter ellipsoid that extends directly downwind of the projected cross-section of the building in the direction of the wind without extending laterally to the sides of the building or vertically above the building similar to the recirculating cavity seen in Figure 6 ("Original Wake"). The lack

of turbulent diffusion results in a model that cannot accurately predict the reduced velocity above the buildings which can still significantly affect the power produced by a wind turbine that is located downwind of buildings.

    To support distributed wind applications, we have developed a new diffusive-wake model for QUIC that extends both laterally from the sides and vertically above the top of the building (see Figure 7 "Diffusive Wake"), using machine-learning techniques on time-averaged high-fidelity LES. A set of equations and their parameters describing the stream-wise and

crosswind components were developed based on comparisons to wind-tunnel data, high-fidelity models, and general understanding of flow characteristics. Parameters for these functions were determined to be themselves functions of



meteorological and building geometry related variables such as atmospheric stability, building dimensions, downwind distance, and relative wind angle. The model was trained against time-averaged data from 72 LES simulations provided by Aeris' using Joint Outdoor Urban-indoor LES (JOULES) model (Bieringer et al. 2021) depicting different building

dimensions, atmospheric stabilities, and wind angles. Several methods of machine learning were used with the data such as data smoothing, non-linear fitting, and genetic programming to calculate parameters for the governing equations. These methods were derived from a suite of python packages including PyVista (The PyVista Developers, 2021), SciPy Optimize (The SciPy community, 2021), Statsmodels (Perktold, Seabold, & Taylor, 2021), and GPlearn (Stephens, 2019). Using SciPy's Optimize curve fitting function, parameter values were calculated as a function of downwind distance for each of the 72 cases.

GPlearn's symbolic regression algorithms where then used on each of the parameters of these 72 cases to generate equations that are functions of atmospheric stability, building dimensions, downwind distance, and wind angle. With these equations and parameters, the new diffusive wake model was implemented as an optional capability in QUIC-URB. The updated QUIC diffusive model could then be run on a subset of EAZ turbines by creating domains using the QUIC Graphical User Interface (QUIC-GUI) and GeoJSON's of building data. Note that all the building object shapes when resolved in QUIC are converted

automatically to rectangular shapes since current functionality of diffusive wake model is limited to rectangular shapes as shown in Figure 6.

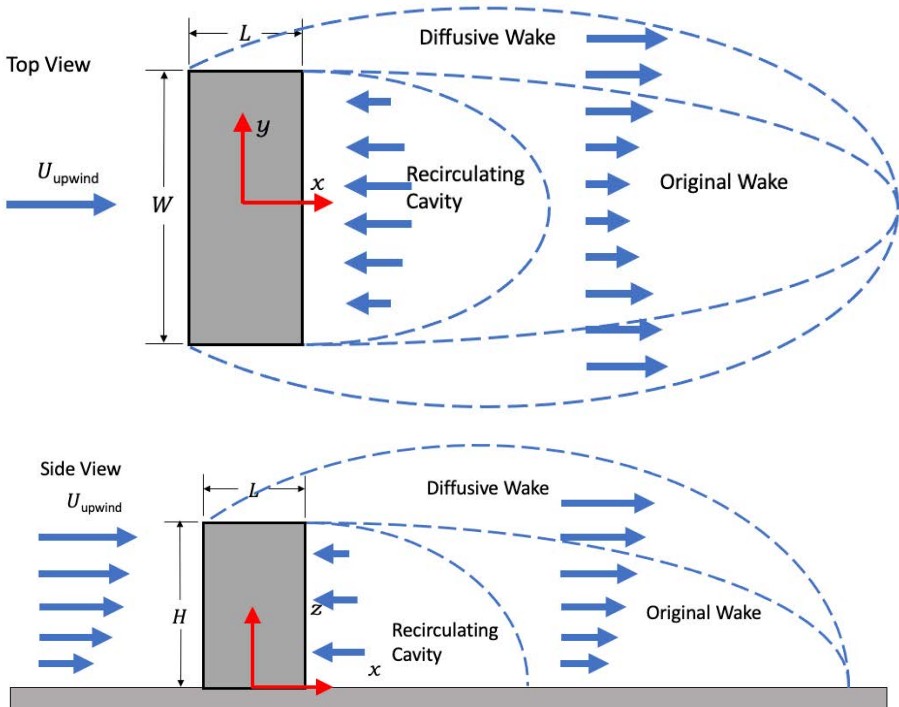

**Figure 6.** Schematic showing the regions where the two wake algorithms, diffusive wake and original wake, as well as the recirculating cavity algorithm, are applied downwind of the building in QUIC.





### 3.4 Data-Driven Methods

The aforementioned models are able to make predictions of velocity deficits given only the inflow wind speed, direction, and a description of obstacles. However, when prior data is available, as it is for this study area, there is an additional opportunity
to take a data-driven approach, replacing or augmenting simulation-derived and wind-tunnel empirical models like those discussed above with statistical or machine learning models fitted to data from the specific site of interest. It is this observation that motivates the development and validation of an entirely data-driven approach to resource estimation. To this end, we utilize observed production data from 307 turbines to train predictive models that are evaluated on the 20 remaining turbines.

Featurization of the obstacles is the first
concern in augmenting the available measurement data so that models may "learn" the impact of obstacles on power generation, or wind speed. To strike a balance between computational complexity and detail, we have developed the featurization
method shown in Figure 7. For each turbine, and each of 36 points along the azimuth, we take inventory of all buildings that fall along a 1-km ray. For the buildings or obstacles along this path, we calculate the number of obstacles ($n$), the maximum
height among obstacles ($maxh$), the distance to the nearest obstacle ($mind$), the total cross-sectional

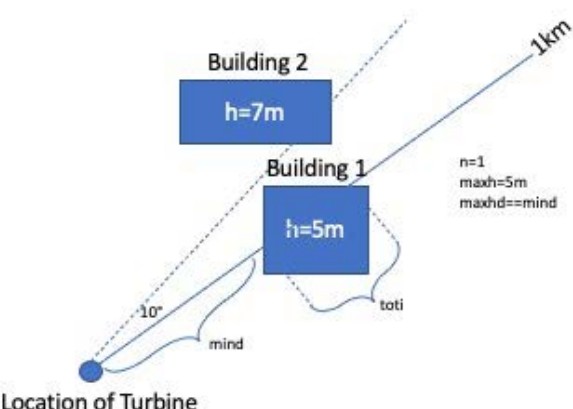

**Figure 7.** Schematic describing featurization. In this example, only one building intersects the 1-km ray at the present angle, so $n=1$ and $maxhd$=mind, while $maxh$=5 (the height of the intersecting building), $mind$ is the distance to the leeward face, and $toti$ is the total length of the intersection in meters.

distance where there are obstacles ($toti$), and the distance to the maximum height obstacle ($maxhd$). Future work may consider deep neural network architectures such as convolutional neural networks (CNNs) that do not require explicit featurization.

To choose an effective modelling approach, we evaluate three different models using these same features: (1) multiple
linear regression (MLR) fitted by ordinary least squares (OLS) (2) random forest (RF) ensemble and (3) support vector regression (SVR) with a polynomial kernel. These models were chosen because they are well known and widely available while providing a range of complexity and modelling approaches. Fitting and hyperparameter tuning is performed with the R statistical computing environment (R Core Team 2020) and caret package (Kuhn 2021). We theorize that SVR may be an appropriate technique for modelling nonlinear relationships given its polynomial kernel, RF may be most suitable for
threshold-based modelling given its underlying tree-based construction, and MLR is useful to provide baseline and easily interpretable results. All models were fitted with and without obstacle features to establish a baseline performance and both predicting power generation (directly) and wind speed (indirectly). Due to the computational complexity of fitting the entire dataset and risk in overfitting, we fit and tune models with 20,000 randomly selected data points, stratified by turbine location and wind direction sector. Experiments using a larger random sample (e.g., 50,000 points) show a small improvement in model





performance (~1%). Fitted performance and variable importance for the three models are provided in Table 2. We find that the RF approach is able to model the training data most harmoniously, while the MLR approach performs admirably given its simplicity. In practice the improvement associated with adding obstacle features is very small (1%).

**Table 2.** Variable importance and performance metrics for fitted models both with and without obstacle features. Results for wind speed (units = m/s) are given. Results from models predicting power generation are similar.

|  | Without Obstacle Features (RMSE; MAE; $R^2$) | With Obstacle Features (RMSE; MAE; $R^2$) | Variable Importance (Ordered most to least) |
|---|---|---|---|
| MLR | 1.04; 0.81; 0.53 | 1.04; 0.81; 0.54 | Reanalysis wind speed, hour, month, maxh, mind (other variables not significant) |
| RF | 0.96; 0.74; 0.58 | **0.95; 0.74; 0.59** | Reanalysis wind speed, hour, sector (wind direction), month, toti, maxhd, mind, maxh |
| SVR | 0.98; 0.75; 0.54 | 0.98; 0.75; 0.54 | Reanalysis wind speed, sector (wind direction), hour, month, maxh, mind, maxhd, toti |

### 3.5 Experimental Design

To evaluate the models described in the three previous subsections, we hold out 20 turbines as a validation set as well as the data from the IEC met tower. The validation set of 20 turbines were chosen from the larger set of 327 because they have significant obstacles in some inflow directions while also representing the geographic diversity and siting complexity in the 285 dataset. Due to the manual nature of some aspects of modelling (in particular, for the QUIC model), and practical feasibility of ensuring accuracy of obstacle descriptions, it was not practical to calculate results for all sites. Hence, we assume this subset of 20 is representative of the larger set. Figure 8 shows two example sites and provides a visual representation of the selection criteria.

In addition to standard error metrics including Root Mean Square Error (RMSE), Mean Absolute Error (MAE) and Mean 290 Bias, for each model and each site we also develop an application-specific measure of error – relative error in annual energy production. This metric is computed as follows:

1. **Hour-Month-Sector (HMS) Matrix:** Calculate the average wind speed for each combination of 10-degree wind direction, hour, and month using model predictions.

2. **Annualized Energy Production (AEP):** Calculate the annual production estimate (MWh) for this site using 295 historical reanalysis data and the HMS matrix as a lookup table.

3. **Relative Error in AEP estimate:** The final metric is computed as the percentage difference between the AP Estimate and the true production of the turbine during the period of study.



To understand the practical performance of models relative to the conditions in which they are applied, each model is evaluated in three specific scenarios/configurations: (1) and (2) with and without added bias correction applied to the inflow
data (Section 2.2) and simple obstacles (3) with enhanced obstacles (Section 2.3).

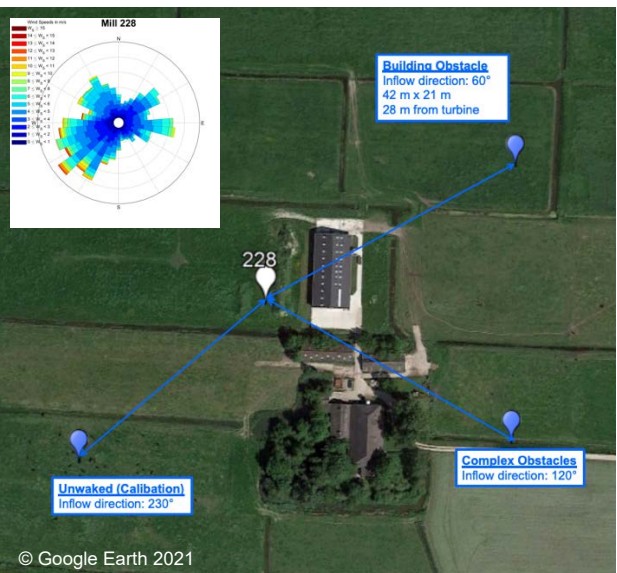
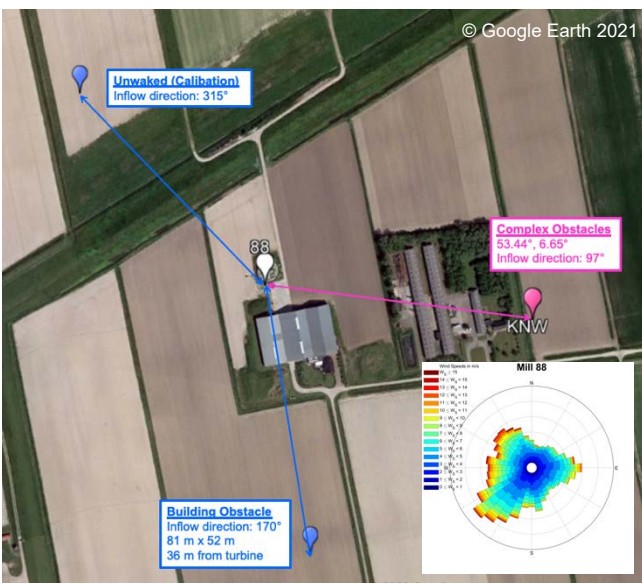

**Figure 8.** Example validation sites (228 and 88) showing typical turbine location as well as the location of obstacles relative to the turbine.

## 4 Results

We subdivide the results below into those that utilize the 8 months of meteorological tower data and those that utilize the validation set of 20 turbines. Due to the inherent strengths and limitations of each performance assessment, we believe they should be taken together to draw a conclusion about practical performance.

### 4.1 Results for IEC Tower

Combined results are provided for the IEC tower site (pictured in Figure 1) in Table 3. While this is only one site, the data is
of high quality for the 8-month study period. We can see that the KNW-Atlas reanalysis data product significantly overestimates the wind speed at this site, resulting in an approximately 13% overestimation in annual production that might be achieved (mean bias 0.25 m/s). We assume that this overestimate is due to the need for obstacle velocity deficit correction. Indeed, when applied, each obstacle correction model reduces the degree of overestimation. However, some models cause too great of an adjustment and skew the results towards the negative. The best performing model overall is the LANL/QUIC
model, achieving <1% relative error (positive bias) in production across the 8 months, followed by Perera (6.4% relative error, positive bias). The best performing model with a negative bias is the data-fitted bias correction model (no obstacles considered)





resulting in average relative error of -8.8%. We believe the data fitted models are less performant here because they were fitted with turbine data rather than met-tower data, and hence are optimized to predict turbine performance.

Figure 9 compares the error process for both the inflow reanalysis data and the QUIC/LANL and ANL methods using a polar plot. As this site has the bulk of obstacles located to the NNW we might expect the error in this direction whereas the other dominant modalities in wind direction (W and SE) appear unobstructed. The plots show that the QUIC/LANL Model makes only modest adjustments to the positive bias in the NNW directions and most significantly addresses overestimates in less-dominant wind flow directions (N and NW). By comparison, the ANL model does better to address the overestimate of wind speeds in the direction of obstacles (NNW) but also exaggerates these effects in some directions (W and NNE). These

results suggest there is significant sensitivity in the choice of obstacle model, in some cases the choice may cause more harm than good in environments where the underlying reanalysis data has relatively low bias, the terrain is simple and turbines can be located to minimize interference from obstacles in the dominant direction of wind flow.

**Table 3.** Results Comparison for IEC Met Tower

| Model | Bias (HMS Average/Point Average) m/s | RMSE (HMS Average/Point Average) | MAE (HMS Average/Point Average) | AEP Relative Error (%) |
|---|---|---|---|---|
| Reanalysis Only | 0.25/0.27 | 1.20/1.28 | 0.88/0.95 | 13.09 |
| ANL (Non-linear) | -0.30/-0.28 | 1.09/1.15 | **0.80/0.86** | -10.53 |
| LANL/QUIC | **0.07/--** | 1.59/-- | 1.21/-- | **0.95** |
| ANL ML (Non-linear) | -0.82/-0.79 | 1.54/1.56 | 1.12/1.14 | -26.42 |
| Data Driven (RF/Wind) | -0.08/-0.10 | **1.05/1.13** | 0.82/0.88 | -10.40 |
| Data Driven (RF/Power) | N/A | N/A | N/A | -9.99 |
| Data Driven (Bias Only) | -0.11/-0.11 | 1.06/1.04 | 0.82/0.88 | -8.81 |
| Perera | **0.08/0.09** | 1.28/1.37 | 0.95/1.02 | 6.36 |
| Perera+ | 0.23/0.25 | 1.20/1.28 | 0.88/0.95 | 12.38 |

**4.2 Results for Turbine Validation Set**

In the next phase of analysis, we calculate similar metrics for the larger dataset of 20 selected validation turbine locations. Table 4 and Figure 10 provide the combined results. We can see that the most performant models across a variety of sites are the data-driven models (NREL ML RF and NREL Turbine-BC, the linear bias correction model). This stands to reason since these models were able to benefit from the rich multiyear dataset of production from turbines in this same environment. Interestingly, the addition of obstacle features adds only a very small (~1%) improvement to the overall accuracy

of this data driven method, suggesting that local bias correction without obstacle models may be sufficient in regions with similar topography and well-sited turbines with few large obstacles. Among those models that operate without *a priori*



information, the LANL/QUIC and ANL Non-linear models both perform well -- 8% and 12% average error respectively,
through the LANL/QUIC model has a smaller spread. The Perera model makes very conservative adjustments to the inflow
reanalysis data. These results again suggest a high degree of sensitivity in model selection with some models overestimating
or underestimating the annual production by as much as 40 or 60% for individual turbines. When existing production data is
available, the best approach for these sites appears to be one driven by prior measurements, achieving 3-5% less mean error.

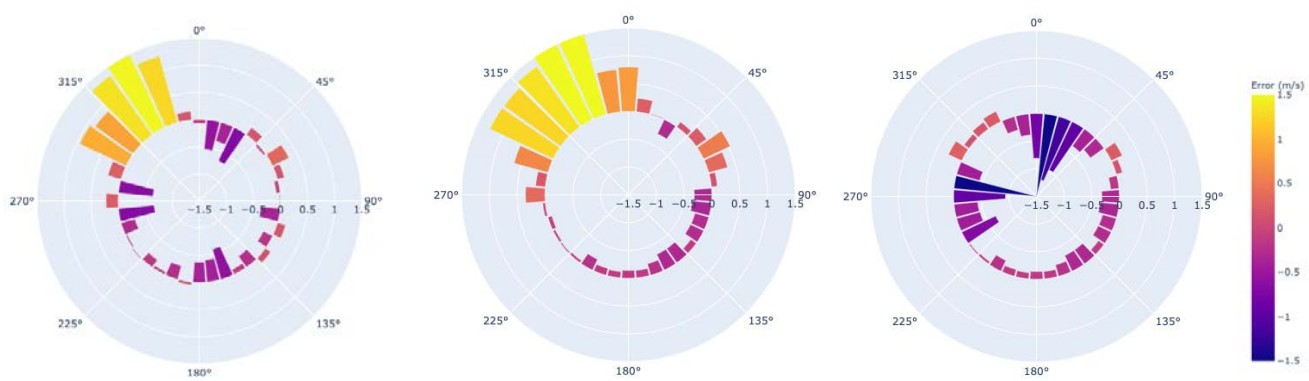

**Figure 9.** Average error in m/s for IEC site grouped by wind speed direction (10 degree sectors) for (a) LANL/QUIC (b) KNW-Atlas
reanalysis and (c) ANL Nonlinear. Positive errors (lighter colours) suggest an overestimate of the wind-resource relative to observed data
345       whereas negative errors (darker colours) indicate the opposite. The scale of errors for each plot are given on the radial access.

**Table 4.** Results Comparison for Validation Sites (averages)

| Model | Mean Bias (m/s) | Mean RMSE (m/s) | Mean MAE (m/s) | Mean AEP Relative Error (%) | Max/Min AEP Relative Error (%) |
|---|---|---|---|---|---|
| Reanalysis Only (15m) | 0.44 | 1.48 | 1.04 | 26.89 | 70.02/-1.66 |
| ANL (Non-linear) | **-0.13** | 1.63 | 1.18 | 6.33 | 55.38/-37.92 |
| LANL/QUIC | -- | -- | -- | 7.96 | 34.05/-16.97 |
| ANL ML (Non-linear) | -0.76 | 2.00 | 1.52 | -11.52 | 43.90/-49.39 |
| Data Driven (RF/Wind) | 0.17 | **1.08** | **0.81** | 4.70 | 36.60/-14.95 |
| Data Driven (RF/Power) | -- | -- | -- | **2.48** | **32.11/-13.31** |
| Data Driven (MLR/Wind) (Without Obstacles) | 0.14 | 1.15 | 0.85 | 5.12 | 40.99/-25.51 |
| Perera | 1.25 | 3.12 | 1.87 | 25.66 | 91.25/4.32 |
| Perera+ | 1.38 | 3.18 | 1.88 | 31.60 | 99.31/9.19 |



**Figure 10.** Box and whisker plots showing performance (median, IQR, 90$^{th}$ percentiles and outliers) for the models studied for annual production estimate relative error (top) and mean pointwise wind speed bias (bottom).

## 4.3 Bias Correction before Obstacle Correction

Being that the data-driven methods perform very well in our analysis, and even without obstacle features, one potential approach would be to combine a data-fitted regional/site level bias correction with subsequent obstacle model (i.e., Perera, LANL's QUIC, or the ANL LOM). In this scenario, the obstacle models would begin with velocity estimates that are closer to observation, perhaps reducing their error but also with the risk of double-counting the effect of obstacles. To understand the practical efficacy of this hybrid approach, we calculated the performance of each model with and without prior bias correction. Figure 11 shows the result of this experiment. We can see that for all models, the addition of prior bias correction significantly



reduces the wind speed estimates, causing a negative mean bias and thereby underestimating the performance of the turbines
in most cases. The Perera model does perform better with bias correction, however this is likely due to the degree to which the
Perera model overestimates the wind velocity rather than as a result of a uniquely positive interaction between these two
models. On the right-hand side of this plot, we can see the performance for bias correction alone, without obstacle assessment,
which tends to perform better (error much closer to zero) than all models except Perera as noted above. Based on these results,
we recommend that either local (data-fitted) bias correction be applied when data is available *or* obstacle correction, when
data is not available, but not both. All models still outperform reanalysis data alone.

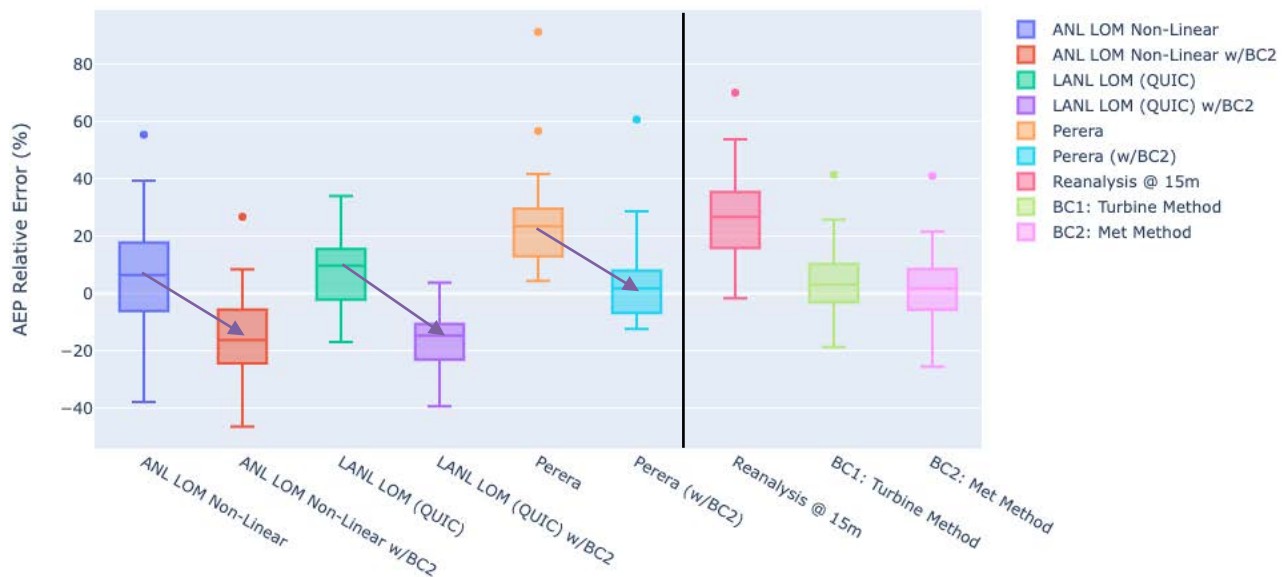

**Figure 11.** Box and whisker plots showing performance (median, IQR, 90th percentiles and outliers, when present) for the models studied, both with and without local bias correction applied prior to obstacle impact assessment. The six boxes on the left of the plot correspond to the models with and without bias correction, while the three on the right are the inflow data and bias corrected data without obstacle assessment.

### 4.4 Quality of Obstacle Data

As a final consideration in the configuration of the models, we have run all models for all selected sites using both the simple
building-only obstacle data available commercially from 3DBuildings.com and with those using our semi-automated
annotation method, utilizing lidar data and manual inspection to add additional buildings and vegetation. Notably not all
models are designed to model vegetation, so for those models that do not explicitly include it (ANL LOM, Perera), we treat
vegetation as if they were solid obstacles of the same size and shape.




Figure 12 shows the result of this experiment. In nearly all cases, adding additional obstacles (whether additional buildings or vegetation) results in generally lower wind speed predictions at the turbine locations. The ANL models overestimate the effect of obstacles in this case, causing an overall negative bias for most sites and thereby underestimating the performance of the turbine. This is a net improvement as the mean error for the simple obstacles is 6% and the performance with added obstacles, -6%. Generally speaking, a small negative error is likely preferable to a positive error for the purpose of performance estimation in siting. Overall, the spread stays high for this model, however, and at some sites the error is greater than 20%. The Diffusive Wake LANL model achieves similar performance with the added obstacles (8% to 8.5% mean error), likely due to a more sophisticated multi-obstacle method and functions designed for the inclusion of vegetation. The Perera model performs somewhat better with more obstacles, but still overestimates wind speeds. The data fitted NREL model performs similarly, having nearly identical mean error characteristics (2.5% to 2.8%) and spread.

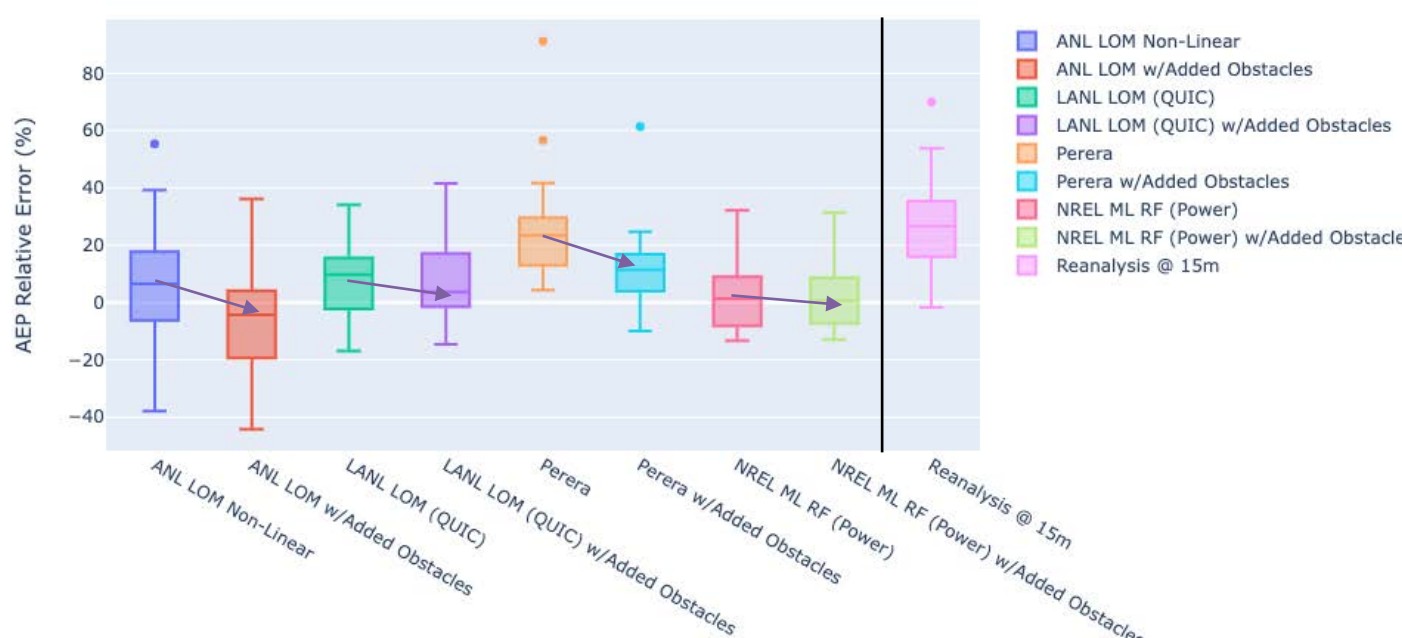

**Figure 12.** Box and whisker plots showing performance (median, IQR, 90[th] percentiles and outliers, when present) for the models studied, both with and without additional obstacles. The eight boxes on the left of the plot correspond to the models with and with and without additional obstacles, while the one on the right shows inflow without obstacle assessment.

These results show the sensitivity of the models studied to the quality of obstacle input data. While generally the results are similar with higher-fidelity obstacle data there are also cases where the added obstacles (or vegetation) may cause a model to underestimate wind speeds significantly. The best-performing models also perform slightly worse in the mean (and slightly better in the median). When it is possible to collect observational data prior to deploying turbines, or when data from



previously deployed turbines is available, it is likely advisable to conduct a data-driven model selection and calibration, thereby determining the best model for a given region and the necessary detail in obstacle data to maximize practical performance. Also, in areas with significant vegetation, it may be most appropriate to consider whether modelling vegetation is necessary.

## 5 Conclusions

This study provides a first of its kind analysis comparing a wide variety of obstacle modelling approaches in the practical
setting of distributed wind resource assessment and small wind turbine siting. We find that mesoscale reanalysis datasets may not adequately consider local topography and obstacles resulting in a significant overestimate in windspeed and performance at typical sites. When possible, data-driven methods fitted to production actuals or measurements from mast-mounted anemometers provide the best opportunity to bias correct the wind resource from mesoscale models, providing the greatest relative benefit. In the absence of site-specific measurements, lower order obstacle models improve estimates at some sites (by
as much as 10 to 15% in annual production estimates), but the resulting accuracy may be sensitively dependent on the choice of model, the quality of the inflow data, and the fidelity of obstacle descriptions. Classical models such as Perera underestimate the impact of obstacles, while in certain circumstances the newer models may overestimate the impact of obstacles resulting in negative bias. This tendency is compounded when combining multiple models (e.g., bias correction and obstacle models) or using higher-fidelity obstacle metadata that includes vegetation. The LANL/QUIC model performs best among the lower
order models studied, however its computational cost is slightly higher than the other models studied and controls on export may hinder broad international adoption.

This study does have several limitations that must be recognized in interpreting the findings:

- The northern Netherlands is an agricultural region, at sea level, with relatively undisturbed inflow wind and very low terrain complexity. To the extent possible, turbine locations have also been chosen to minimize impact from obstacles.
Hence, our results are likely limited to those in similar environments and should not be applied in widely different settings. In future work we plan to continue these studies in additional settings such as moderately built environments in the central US, as well as environments with more terrain complexity such as the US Mountain West.

- While we do make use of anemometer measurements for IEC site, the bulk of our dataset is derived from turbine production data which has inherent limits in both accuracy and the ability to infer wind speed *ex post facto*.
Measurements from production wind turbines necessarily limit the impact of obstacles because siting decisions have been made to avoid them. Notably, however, we do find broad agreement in the conclusions from both anemometer and production-inferred datasets. Nevertheless, these results should be reconfirmed when additional meteorological data are available.

- We study a range of models from those just introduced, to those introduced more than 30 years ago. The most recent
models are still being actively developed and improved and it is entirely possible that future versions may perform better or differently.



In summary, based on the conclusions of this study, we recommend that small wind operators take care when accounting for obstacles and whenever possible the greatest benefit is likely to be realised through measurement campaigns and careful use of existing mesoscale data products, ideally coupled with bias correction. Small wind operators may wish to consult multiple

data products and techniques in their analysis in order to determine a range of possible results instead of just one. When *a priori* obstacle impacts are needed, without pre-existing data, modern lower order models are likely to reduce error as compared to classical models such as Perera. As compared to data-fitted models, analytical models may find additional value in micro-siting applications where the exact velocity deficit due to obstacles is not needed, but rather the shape and extent of their impacts. In future work, we expect to extend and improve upon these results by considering additional environments and

applications.



**Code Availability**

Software for the developed models is currently being prepared for release through our project website (under development) and API (https://dw-tap.nrel.gov/).

**Data Availability**

Data used in this study is either already available publicly (e.g., KNW Atlas reanalysis data, AHN3 lidar data) or proprietary (e.g., EAZ turbine production data). We do not expect to release additional supporting data with this publication.

**Author Contributions**

Concept, manuscript preparation, software development and execution were performed by C. Phillips. Validation methodology was developed by L. Sheridan. Models were developed by C. Phillips, P. Conry, D. Fytanidis, M. Nelson, and N. Duboc. Obstacle mapping methodology developed by S. Zisman and D. Duplyakin. Interpolation methodology and evaluation by D. Duplyakin. All authors contributed to manuscript edits and technical review and interpretation of results.

**Competing Interests**

The authors have no competing interests to report.



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
