# Peer review of "Evaluation of Obstacle Modelling Approaches for Resource Assessment and Small Wind Turbine Siting: Case Study in the Northern Netherlands"

_Wind Energy Science, 2021_

## Author Comment (AC2)

Dear referees, editor, and community,

Thank you for your time spent reviewing and providing careful consideration of our manuscript. Please find detailed responses to the comments and requests below. Our responses are in orange (in the attached PDF) and indented with > angle brackets alongside the reviewer comments, included for context (both in the text response and the PDF). In addition to the requests from the referrees we have also made edits to the references to match the WES format (as requested by the WES editorial staff), and added an acknowledgments section (as required by the authors' institutions).

**RC1**

The authors performed a comparative study of modelling approaches for the interaction of wind turbines with nearby obstacles in siting analyses. Four different strategies were tested: the classical Perera model, a novel analytical model derived from an extensive dataset of CFD simulations (ANL), a modified urban dispersion model (QUIC-URB) and a machine-learning model, fitted to site-specific data.  As benchmark for the evaluation of the models 'performance, wind data derived from production data of a set of small turbines located in the Netherlands were employed. The analysis gives useful insights into the capabilities and limitations of the different models and how they affect the predicted production of the site.

The reviewer believes that the topic and the activity are very interesting, innovative and worthy of investigation. The approach is rigorous and consistent throughout the whole activity. The paper is well presented, and the results are clear.

> Thank you very much for your comments and encouragement.

Some specific considerations:

- The use of first-person plural is not recommended for a scientific publication;

> Sure, we are happy to make this change.

- line 49: it is not clear which models are from the literature and which from the authors. Please clarify this aspect;

> Yes, thank you, this has been fixed.

- line 97: references to "prior validation results" are missing;

> Thank you, the reference to (Stepek 2015) has been moved to make this clearer.

- line 100: it is not clear from the text that the work is from some of the authors;

> Good point, this has been clarified.

- line 124: in the Reviewer's opinion, the term "layers" would be more fitting than "slices" in this context;

> We replaced with "layers of this data frame"

- Section 3: it would be useful to add a table comparing the different models in terms of main characteristics, type, computational effort, input data, etc...;

> Yes, thank you, this is a good suggestion and we will add this as Table 2.

- Typos: line 99, "establishedd" is written with two d; line 296, "AP" instead of "AEP";

> Fixed, thank you.

The Reviewer recommends the publication of this paper after the proposed minor modifications have been performed.

RC2

Dearest Authors,

The paper addresses in detail one of the most important barriers for small wind development, firstly the common lack of observational data and secondly the lack of accurate, cost-competitive, and friendly wind resource assessment tools for small wind applications. The paper contains a comparative analysis of a comprehensive number of methods for wind resource assessment of sites with obstacles based on different data sources.  All the methods analyzed are quite well explained but they are obviously very sensitive to the quality of obstacles description and input data.

> Thank you.

The most relevant requirement nowadays is to get rid of the use of high computationally time-consuming tools based on HF CFD models but keep the accuracy in a wide range of applications. In this way, the main contributions of the paper, apart from the comparative multi analysis, are the development of AI (AN /ML)-accelerated CFD tools and the adaptation of friendly commercial urban dispersion models to small wind turbine site assessment applications. Both solutions offer time to results significantly shorter allowing a faster analysis with reduced cost and error for the specific conditions of the trial test. (Flat terrain, limited obstacles and similar 300 small wind turbines with available certified power curve) but it seems to be difficult to replicate this procedure in other sites with different conditions (for instance complex terrain, significant obstacles and/or different level of wakes, blockage, etc.).

Regarding the type of obstacles included in the analysis, only buildings and significant vegetation (trees) have been included, but there are other kinds of obstacles like fences, walls, very common that also plainly affect the wind inflow in small wind applications.

The results obtained are clearly represented by standard error metrics but I would like to highlight the contribution of an application-specific measure of error as the relative error in annual energy production. I consider that these new metrics proposed are quite useful for actual small wind applications.

Finally, just to highlight the significant influence, in the performance of the different models, of the relative situation of the obstacles, the wind turbine, and wind flow directions even using IEC Met tower or turbine validation data sets. The hybrid approach to combine a data-fitted site-level bias correction with a subsequent obstacle model is really interesting to reduce the uncertainty in the performance of the models.

In conclusion, the content of this paper is considered very appropriate and relevant research for DW applications. The conclusions are well-reported and supported by the results obtained, the tables and figures included are consistent.

> Thank you – we appreciate your thoughtful review and your experienced perspective.

RC3

The authors provide a comparative evaluation of different ways to model the effect of obstacles on the performance of small wind turbines. The models considered are of varying degrees of fidelity.

The manuscript is well-written and the subject is scientifically relevant. The conclusions are clear and include a discussion of model limitations. I believe the manuscript is of the level expected for a paper in WES. However, I have some comments and questions that I should like to see addressed in the final version.

> Thank you.

One of the approaches considered involves low-order models trained on RANS simulations. I found it difficult to assess the quality of these simulations, as the reference (Fytanidis et al. 2021a) pointing to those simulations does not appear to be readily available. Some more information on the CFD simulations should be added to the paper, allowing to assess the quality of the CFD simulations. A clarification on the choice of a less common turbulence model would also be welcome.

> Thank you, yes we will add additional detail here.

The resolution of some of the figures should be improved. In particular, the labels on most figures don't look good at all. Saving the original in a more appropriate format is probably enough to fix this. Also the gray-blue box on a grey background adopted for figs 10-12 may not be the best choice, and definitely has a very low data-ink ratio.

> Good point, we have: For figure (1) recomposed the subfigure so the previously tiny axes labels are no longer needed; figure (2) has been regenerated in a different format with better axis labels; figure (3) has been regenerated with larger labels for the turbines; figure (4) unchanged; figure (5) has been rendered larger to allow better visibility; figure (6) unchanged; figure (7) unchanged; figure (8) recomposed the subfigure with simplified axis labels; figure (9-12) unchanged. Though this retains the grey backgrounds on figures 10-12, we believe it improves readability. We hope these changes address your concerns.

The RMSE should be defined more explicitly, preferably with an equation, to clarify what is being compared to what, for which record length.

> Absolutely – we have improved the description of these metrics.

For the IEC site it was not clear to me what information was included in the training of the data-driven model. I would think the available information is too limited, but perhaps I am mistaken. Could this be clarified?

> Upon review, we agree that the first paragraph of section 4.1 is confusing in this regard, we edited to clarify.

Minor comment:

- the spelling of Amsterdam airport is Schiphol

> Fixed. Thank you.